# Evolutionary conservation of within-family biodiversity patterns

Paola Laiolo[1] ✉, Joaquina Pato[1], Borja Jiménez-Alfaro[1] & José Ramón Obeso[1]

The tendency for species to retain their ancestral biological properties has been widely demonstrated, but the effect of phylogenetic constraints when progressing from species to ensemble-level properties requires further assessment. Here we test whether community-level patterns (environmental shifts in local species richness and turnover) are phylogenetically conserved, assessing whether their similarity across different families of lichens, insects, and birds is dictated by the relatedness of these families. We show a significant phylogenetic signal in the shape of the species richness-elevation curve and the decay of community similarity with elevation: closely related families share community patterns within the three major taxa. Phylogenetic influences are partly explained by similarities among families in conserved traits defining body plan and interactions, implying a scaling of phylogenetic effects from the organismal to the community level. Consequently, the phylogenetic signal in community-level patterns informs about how the historical legacy of a taxon and shared responses among related taxa to similar environments contribute to community assembly and diversity patterns.

[1] Research Unit of Biodiversity (CSIC, UO, PA), Oviedo University, Mieres, Spain. ✉email: paola.laiolo@csic.es

As a result of the process of descent with modification, evolutionarily related organisms tend to possess more similar biological features than distantly related ones[1]. This legacy of common ancestry, termed phylogenetic signal, has long been recognised in physiological mechanisms, life histories, body plan and, more recently, in interactions[2–4]. This pattern can then scale to higher levels than aggregated individual traits: differences in range size among species[5,6], or in speciation-extinction dynamics among clades[7], may vary non-randomly with respect to phylogeny. These features have a profound influence on species richness patterns. They dictate the size of global species pools, the environmental conditions the members of a clade can tolerate, the regions they can colonise and the intrinsic limitations to their dispersal[8–10]. The question thus arises whether the conservatism of these features, influencing which species occur in a given place, scales up to ensemble-level properties, influencing how many species of a clade can coexist.

At a global scale, species richness gradients tend to match the spatial variation in measures of energy and productivity in most clades, irrespective of their phylogenetic relationships[11–13]. Conversely, elevational and other environmental clines that affect the community level are often contingent on the type of organism, its natural history and environment[14–17]. The idea that these small-scale patterns may be phylogenetically constrained has emerged from observations of repeated, clade-specific, community patterns in different regions, even in the absence of shared species[18–21]. Assessing the generality of this phenomenon requires comparative analyses on multiple clades, which to our knowledge has not been done at the scale of ecological communities. In fact, multi-taxa studies often treat the diversity responses of clades as independent replicates for ecological or biogeographic tests[15–17]. Yet, independence cannot be taken for granted, as functional and taxonomic boundaries for delimiting communities often overlap. Phylogenetically close species are to some extent functionally similar, and thus more likely to interact and, in turn, form communities[22]. Such communities ("taxocoenoses") display their own evolved interactions and tolerances[23,24], as well as specific community patterns. These patterns, unlike global ones, emerge from species originating and evolving elsewhere, and coexisting and replacing each other in response to local environmental filters.

Here, we investigate whether there is a fundamental phylogenetic pattern at the heart of the differences in community diversity responses among higher clades, and whether these responses are grounded in the phylogenetic conservatism of organismal or higher-level traits. To assess to what extent diversity–environment relationships diverge over evolutionary time, and the generality of this phenomenon, we compare patterns of communities formed by members of the same family (confamilial communities) within major taxa—lichen, insects and birds. Along a common regional elevation gradient, we analysed the phylogenetic signal in four fundamental diversity–environment relationships: the elevational gradient of community species richness, and the turnover of species with geographic distance, elevational distance and habitat dissimilarity[25,26]. Our response variables, hereafter diversity variables, describe the behaviour of confamilial communities in these environmental and geographic contexts (Fig. 1). We test the prediction that these variables retain a significant phylogenetic signal, quantifying their degree of covariation with family-level phylogenies within the three major taxa. Then, we centre on a set of organismal or higher-level features that are important in community assembly (Fig. 1) and test whether they display a significant phylogenetic signal. If the phylogenetic pattern in diversity responses emerges from these features, we expect that they are still conserved at the family-level and achieve the same

importance of phylogenetic relationships in explaining differences among families at the community level. In particular, we expect that life histories, body plans and the incidence of interactions may correlate with diversity shifts along elevation or habitat, for their influence on physiology and ecological tolerances[27,28], and in turn on the ability of species to establish and persist[14]. We also expect that the incidence of traits linked to dispersal[29] may be more strongly associated with differences in species colonisation ability and, together with regional distributions and species richness[30], may contribute to the spatial replacement of species (Fig. 1).

We report a correspondence between community-level patterns along elevation, certain organismal features, and the relatedness of the families that produce the patterns, which suggest that phylogenetically conserved properties of organisms translate effectively into the structuring of entire ecological communities.

## Results

**Phylogenetic signal in community patterns**. The biogeographic setting of this study is the steep elevation gradient of the Cantabrian Mountains in northern Spain, where fourteen lichen, nine insect and nine avian families are species-rich and widespread enough to study their community patterns (Supplementary Fig. 1). Families represent guilds of species with similar body plans and ecology (e.g., woodpeckers, jelly lichens, grasshoppers, etc.) separated by deep evolutionary times of tens (bird families) or hundreds (lichen and insect families) of millions of years (Supplementary Fig. 1). These families constitute a random subset, with respect to phylogeny, of the regional pools of each major taxa (D statistic for phylogenetic structure = 0.0–1.6, all $P$-values > 0.09).

Within each major taxa, we used two methods to quantify the phylogenetic signal in α- and β-diversity variables of families (Fig. 1). We computed a phylogenetic signal statistic, Blomberg $K$, and performed regressions on distance matrices in which pairwise dissimilarities between families were regressed on divergence times (the time since these lineages split). Both methods highlighted significant phylogenetic patterns in the differences among families regarding the elevational gradient of diversity and turnover. According to both methods, $Edf$-α of all major taxa, $Peak$-α of lichens and insects, $Slope_z$ of birds and $R^2_z$ of insects diverged significantly over evolutionary time (Fig. 2 and Supplementary Table 1). Another consistent result was the phylogenetic signal in the steepness of insect habitat turnover (Supplementary Table 1). Phylogenetic relationships explained from 6 to 41% of the variance between families in these diversity variables, and significant $K$-values approached 1 in most cases (Supplementary Table 1). These results were obtained despite the low power of tests comparing phylogeny-based vs. null models in the relatively small sample of families. The distribution of likelihood ratio statistics δ of simulated data sets under a null model with random draws independent of the phylogeny, and under a simple model of stochastic evolution (Brownian motion BM model), exemplify this low power, with wide overlapping areas between models in δ density distributions (Fig. 2 and Supplementary Fig. 2). Yet, when these models were fit to our data set, they confirmed previous results for the above-mentioned variables. BM models better fitted data than null models but observed δs were slightly greater than expected BM averages, indicating that we could detect only strong signals (Fig. 3 and Supplementary Fig. 2). On these variables we centred further analyses.

**Mechanisms beyond the conservatism of diversity patterns**. Both Blomberg statistics and regressions on distance matrices

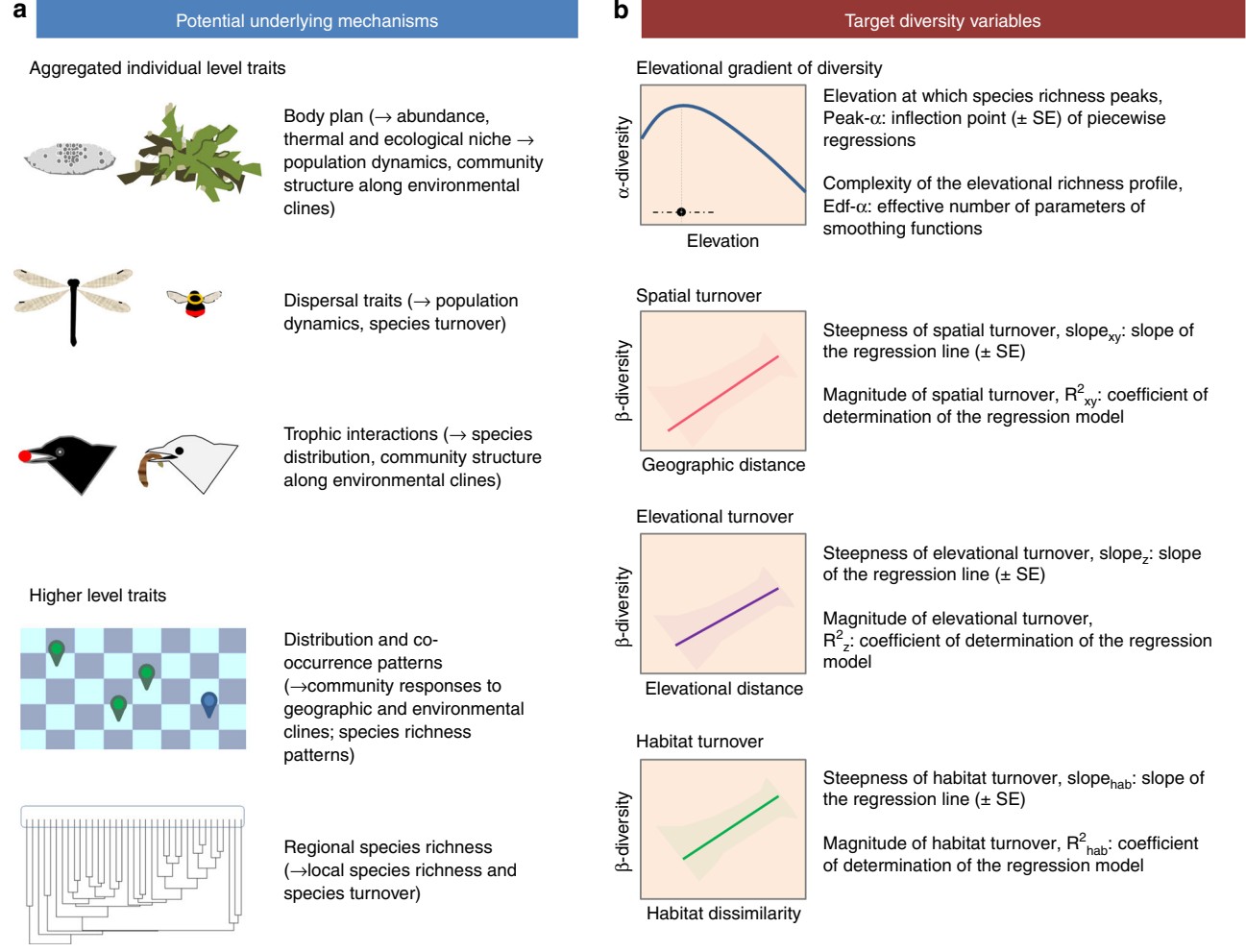

**Fig. 1 Graphical representation of diversity–environment relationships and the organismal and higher-level features that may influence them. a** The intrinsic features and the mechanisms through which these features might influence diversity variables. **b** The eight target responses (diversity variables) estimated for each of the four diversity–environment relationships and confamilial community.

highlighted a significant ($P < 0.05$) phylogenetic pattern in the body size of birds (Blomberg $K = 1.37$; regressions on distance matrices: $R^2 = 0.12$) and insects ($K = 1.96$; $R^2 = 0.30$) and the frequency of small endolithic forms of lichens ($K = 1.69$; $R^2 = 0.44$) (Fig. 2 and Supplementary Table 1). More closely related families also included similar frequencies of species interacting with plants (in the case of animals), or with green algae vs. cyanobacteria (in lichens) (Blomberg statistics: $1.31 < K < 2.05$, $P < 0.05$) (Fig. 2 and Supplementary Table 1). The distribution of $\delta$ values for these organismal features suggests no support for the null model, but also low support for a simple evolutionary model such as the BM model (Fig. 2). Higher-level features had no relationship to family phylogeny: evolutionary history did not help explain why families were rare or widespread, species-rich or -poor, or co-occurred (Supplementary Table 1).

Some organismal features explained variation in the above-mentioned diversity variables as likely as phylogeny when the performance of models with or without a phylogenetic correlation structure was compared (we assumed equal performance when $\Delta AICc < 3$, see Table 1). In particular, it seems that insect and avian families with more species depending on plant resources had more complex elevational diversity clines (higher *Edf-α*) than families of secondary consumers (Fig. 3; Supplementary Fig. 4; and Supplementary Tables 4 and 5). In lichens, it was the incidence of symbiosis with green algae that was associated with

complex elevational diversity patterns (Fig. 3; Supplementary Fig. 4; Supplementary Tables 4 and 5). Families with higher elevation diversity-peaks had greater incidence of endolithic forms in lichens and reduced wing length in insects. In the latter taxon body size also correlated with *Peak-α* and *Slope*hab. Models including evolutionarily labile features (regional species richness and distribution) as predictors had low performance ($\Delta AICc > 5$); only wing length received a certain level of support for predicting bird *Edf-α* (Table 1).

## Discussion
A recurrent view of community ecology is that of a discipline in which generalisation is rarely achieved[31,32]. This study provides evidence of the difficulties of dealing with emergent patterns in complex ecological systems—the same environmental gradient engenders diversity patterns that diverge among families (Fig. 2) and correlate with variation at a different hierarchical level (Fig. 3). In spite of this, a general natural constraint —phylogenetic relationships—explains 6–41% of the observed differences among clades in some of the most widely documented biodiversity patterns. The shape of the diversity-elevation trend, the location of diversity-peaks along elevation, and elevational species turnover, displayed the most consistent phylogenetic signal, with robust results among statistical approaches and the (disparate)

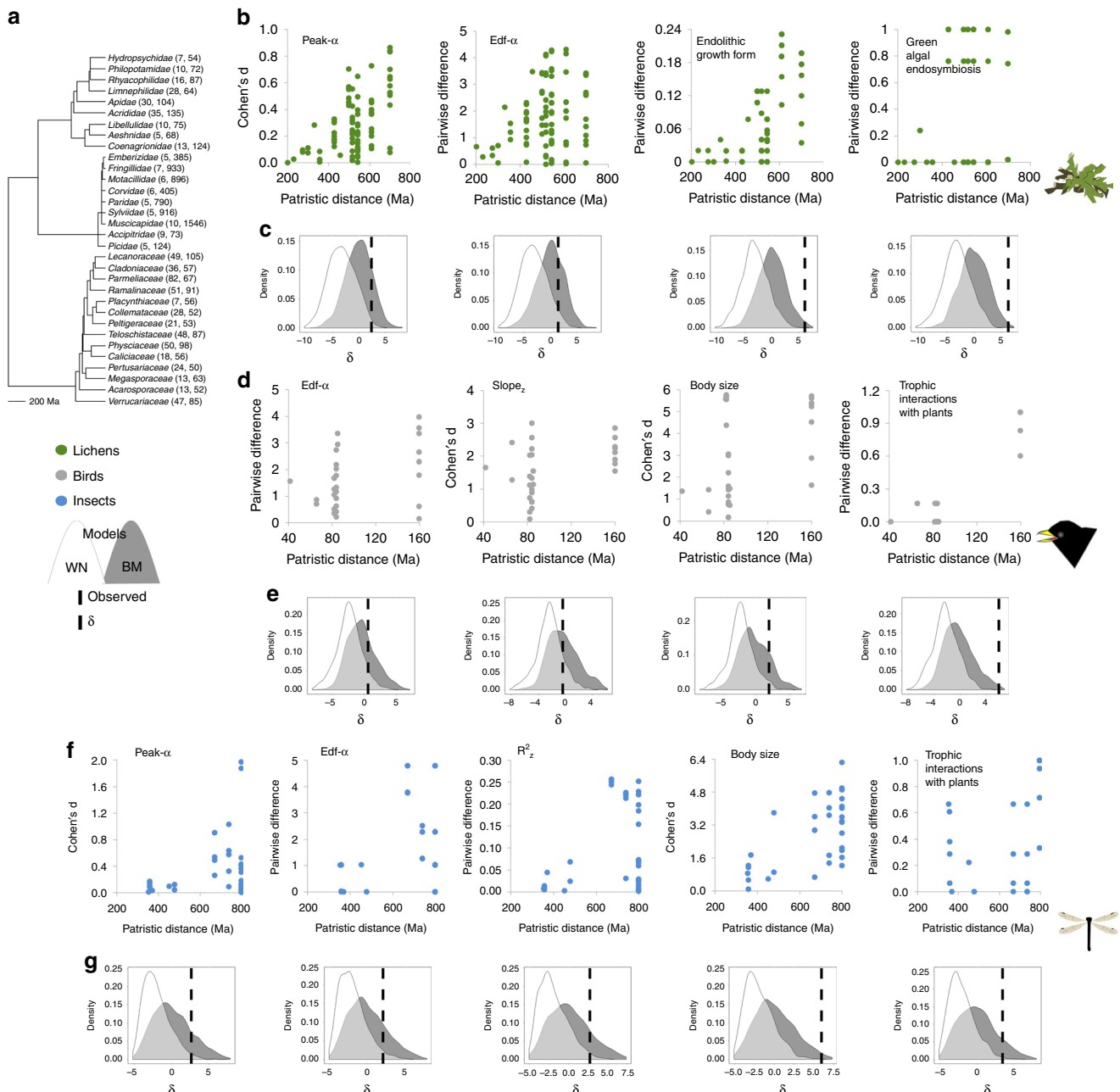

**Fig. 2 Significant phylogenetic patterns in diversity responses and organismal traits. a** Time-calibrated phylogenetic tree of families, with the number of species and sampling plots in parentheses (the three major taxa are represented in a single tree for representative purposes only). **b, d, f** Scatterplots representing how dissimilarities in community responses or organismal traits, quantified as effect sizes or raw differences on the *y*-axis, increase with evolutionary time, represented by patristic distances in million years (Ma) in the *x*-axis; each point represents a pairwise comparison between families. **c, e, g** The density plots below each scatterplot represent, for the same variables as scatterplots, the distribution of the likelihood ratio statistics $\delta$ for a two model comparison, a null model with random draws independent of the phylogeny (white noise WN model, lighter density distributions) and a Brownian motion BM model of evolution (darker density distributions). The dashed vertical line indicates the observed value of $\delta$ when the models are fit to our data set. A total of 1000 replicates were used for each distribution. Peak-$\alpha$: elevation at which species richness peaks. Edf-$\alpha$ effective number of parameters of the smoothing function, Slope$_z$ elevational turnover, $R^2_z$ magnitude of elevational turnover. Source data are provided as a Source Data file.

organisms we studied. The association between these diversity responses and phylogenetic relationships took the form of a progressive increase in both dissimilarity of responses and its variability with evolutionary time. The drift (scatter in Fig. 2) of community responses at deep evolutionary time, apparently matching a Brownian motion model of evolution (Fig. 2), is in line with the stochastic element predicted by neutral theories of biodiversity[33]. The decline of similarity in certain interactions

and body plans in distantly related taxa is another result compatible with, in this case, macroevolutionary theories. The breakdown of phylogenetic inertia and the origin of evolutionary innovations is generally placed at these expanded timeframes (e.g., family-level)[34,35].

Variation in aspects of the ecological niche was significantly associated with changes in local species richness along elevation among families. The latitudinal gradient of species richness has

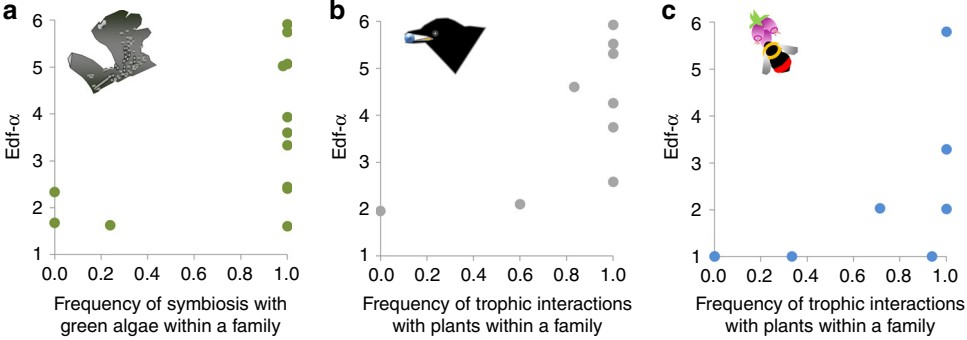

**Fig. 3 The complexity of species richness-elevation relationships increases for certain types of interactions. a, b, c** Scatterplots of the relationships between the parameters of the smoothing function Edf-α of confamilial communities, on the y-axis, and measures of the incidences of green algal endosymbiosis in lichens and of trophic interactions with plants in animals, on the x-axis. Each point represents a family. Source data are provided as a Source Data file.

also been explained by hypotheses based on phylogenetic niche conservatism within families[9]. Yet, the diversity-latitude slopes of families (e.g., among families) appear to be evolutionarily labile, a pattern attributed, among other things, to the general positive effect of energy availability on clade diversification rate[13,36] and the tendency of clade members to maintain their ancestral ranges[9]. Elevational gradients clearly do not reproduce this phenomenon at smaller scales, as most of the species that make up communities in mountains have not originated in situ along these gradients[15,37,38]. Although cold temperatures limit environmental carrying capacity and, thus, the species richness of most taxa at high elevations, this constraint to population growth is often restricted to the upper limits of the gradient[15,39]. In these conditions, and in the absence of local gradients in diversification rate comparable to latitudinal ones, niche traits and their constraints to variation may assume greater relevance in community assembly than variation in higher-level features[40]. In support of this, we found that features such as regional species richness or distribution played no role in the above-mentioned community-level differences among clades, and these features were evolutionarily labile. Conversely, some features at lower levels, defining body plan and interactions, explained variation in some diversity variables even when phylogenetic structure was not accounted for (Table 1), suggesting a mechanistic explanation for the phylogenetic patterns in diversity responses.

Both neontologic and palaeontologic studies have shown that clades diversify in morphospace early in their histories, generally while at relatively low-taxonomic diversities[34]. Affinities in body plan may then lead to similar climatic requirements, i.e., niche conservatism for temperature optimums and distributions[9]. In mountain regions, this conservatism has likely contributed to moving pre-existing species in broadly similar directions during glacial-interglacial cycles, driving present-day elevational profiles of species richness[24]. In line with this, we found an essential connection between organism and community patterns in lichens and insects: diversity peaked at higher elevations in families of endolithic and short-winged forms of lichens and insects, respectively. Both features are generally associated with life in harsh/windy environments[27,41], and may thus favour the settlement of species bearing these characteristics in highlands. We also found a deep phylogenetic constraint in the incidence of interactions with plants in animals and type of photobiont in lichens, in line with results obtained with a finer categorisation of these interactions by other studies[42–45]. These interactions appear to be associated with the complexity of diversity-elevation relationships, which increases with the incidence of green algal photobionts in lichens, and of plant resource/host in animals (Fig. 3). A possible explanation for this relationship is that the

distribution of these partners[46,47] or resources/hosts[48] is influenced by factors other than climate, e.g., changes in habitat types, topography or substrates, which do not display uniform variation along mountainsides[15]. If the distributional limits of species track those of their interactors[48], community patterns of greater complexity may also emerge, an idea that should be explored in other foodwebs.

Our findings have important evolutionary and ecological implications. The scaling of phylogenetic signal from the organism to community level points to an emergent homology in ensemble-level features. The observed similarity over tens or even hundreds of millions of years challenges the view of evolutionary lability of ecological features[49], pointing to community trends that may persist within a lineage over long (macroevolutionary) times. Results also challenge basic expectations of multi-taxa ecological studies—that responses across common gradients should converge under similar environmental pressures—at the same time identifying new opportunities for research in this field. The development of hypotheses based on natural history, phylogenetic patterns and the variability of study systems is a way to increase the generality of results obtained across multiple taxa. The dynamic relationship between autotroph interactors and heterotroph diversity patterns described here suggests another potential approach to the study of diversity–environment relationships, targeting bottom–up effects that propagate through foodwebs and communities. This is just a reduced set of potential outcomes of our results in various fields concerned with biological diversity. We hope that this study will spur increasingly refined analyses, to uncover present-day community patterns and generate rigorous hypotheses on their evolutionary imprint.

## Methods

**Data sets and study system**. With data that we collected or obtained from the literature (Supplementary Note 1), we built three presence-absence site × species matrices from surveys conducted in the Cantabrian Mountains—one with 122 sites and 726 lichen species, a second with 2347 sites and 114 bird species, and a third with 530 sites and 233 insect species (Supplementary Fig. 1). Geographic coordinates, elevation and the proportion of habitat types completed the information about the sites. We chose the taxonomic resolution of the family as the unit of analysis, and selected those families inhabiting ≥50 plots and including ≥5 species to study community patterns. Families not only correspond to well-defined monophyletic and functional groups[50–52], but also represent a common unit of analysis for phylogenetically comparative studies of diversity patterns among taxa[11–13]. We discarded lower resolutions (i.e., genera) because many of these taxa lack monophyly (e.g., *Chorthippus*[53]) or functional distinctiveness (e.g., tits *Parus* vs. titmice *Poecile*[54]). We also discarded higher resolutions (e.g., orders), as they were insufficient for analyses.

The phylogenetic relationships among families were derived from the TimeTree data set[55] as the median node times calculated from available literature. Node ages were used to build a pairwise matrix of patristic distance (twice the node age) among families (Supplementary Table 2). This phylogenetic inference matched that

**Table 1 List of generalised least square regression models testing for the influence of organismal features on those diversity responses with strong evidence of phylogenetic signal.**

| | AICc weight | ΔAICc |
|---|---|---|
| **Lichens: peak-α** | | |
| Endolithic (2912*) + Algal symbiosis; BM | 0.84 | 0.0 |
| Endolithic (2706*) + Algal symbiosis | 0.14 | 3.5 |
| **Lichens: Edf-α** | | |
| Endolithic + Algal symbiosis; BM | 0.29 | 0.0 |
| Endolithic; BM | 0.23 | 0.6 |
| Algal symbiosis; BM | 0.14 | 1.5 |
| **Endolithic + Algal symbiosis (2.11·)** | 0.12 | 1.8 |
| **Algal symbiosis (2.10*)** | 0.07 | 2.8 |
| BM | 0.07 | 2.9 |
| Endolithic | 0.05 | 3.5 |
| **Insects: peak-α** | | |
| **Size-corrected wing length (−1600***) + Plant Interaction (1816**)** | 0.48 | 0.0 |
| Size-corrected wing length (−2099*) + Plant Interaction (1849***); BM | 0.37 | 0.5 |
| **Body size (1687*) + Plant Interaction (1842***)** | 0.11 | 2.9 |
| Body size + Plant Interaction (1755***); BM | 0.04 | 5.0 |
| **Insects: Edf-α** | | |
| Plant interaction (2.24·); BM | 0.24 | 0.0 |
| Size-corrected wing length + Plant Interaction (2.99*); BM | 0.18 | 0.6 |
| BM | 0.16 | 0.7 |
| Size-corrected wing length; BM | 0.08 | 1.2 |
| Body size; BM | 0.08 | 2.1 |
| **Size-corrected wing length (−3.03·) + Plant Interaction (3.83*)** | 0.07 | 2.4 |
| **Plant Interaction (2.17·)** | 0.07 | 2.5 |
| **Size-corrected wing length** | 0.02 | 4.7 |
| **Insects: $R^2_z$** | | |
| BM | 0.84 | 0.0 |
| Size-corrected wing length; BM | 0.06 | 5.1 |
| **Insects: Slope$_{hab}$** | | |
| BM | 0.76 | 0.0 |
| **Body size (0.11*)** | 0.18 | 2.9 |
| **Birds: Edf-α** | | |
| **Plant interaction (2.95*)** | 0.24 | 0.0 |
| BM | 0.19 | 0.4 |
| Plant Interaction; BM | 0.18 | 0.5 |
| Size-corrected wing length (−1.05*); BM | 0.14 | 1.1 |
| **Size-corrected wing length (−0.89·)** | 0.10 | 1.5 |
| Body size | 0.05 | 3.3 |
| Body size; BM | 0.04 | 3.3 |
| Body size + Plant Interaction; BM | 0.02 | 4.7 |
| **Birds: slope$_z$** | | |
| BM | 1.00 | 0.0 |

Models were ranked on the basis of ΔAICc, the difference in AICc points from the best model, and AICc weight, the probability of a model to be the best one. Models may or may not include a phylogenetic structure based on the Brownian motion BM model of evolution. When "BM" appears in the list of predictors, it means that phylogeny is affecting the relationship between organismal features and diversity variables; when "BM" appears alone, the model includes the sole intercept and phylogenetic structure. Estimates of significant regression coefficients are shown in parentheses.Only models with ΔAICc < 5 are shown.
Best supported models (ΔAICc < 3) with no phylogenetic structure are indicated in bold.
Peak-α elevation at which species richness peaks, Edf-α effective number of parameters of the smoothing function, Slope$_{hab}$ habitat turnover, Slope$_z$ elevational turnover, $R^2_z$ magnitude of elevational turnover.
***P < 0.001, **P < 0.01, *P < 0.05, ·P < 0.10, generalised least square regressions. Sample size corresponds to the number of families, 14 of lichens, 9 of insects, 9 of birds.

fructicose macrolichens, and reproduction modes in order of increasing propagule size and decreasing mobility, from long-dispersing sexual spores to thallus fragmentation. We obtained information on the diet of animal species and then categorised trophic interactions in broad non-exclusive categories for comparative purposes: those with plants (including pollinators, herbivores, frugivores, and granivores), with invertebrates, and with vertebrates (the latter only for birds). We also compiled a data set of the type of autotroph endosymbiont(s) in lichen species, and again clumped this information in two broad categories of interactions: those with cyanobacteria and those with green algae. For each phenotypic trait and each animal family, we obtained the mean and standard error of traits of local member species, using log-transformed values of body size. In the case of interactions, we calculated the average incidence, per family, of species with each type of interaction. In lichens, in which all functional aspects were expressed in terms of presence/absence of a given feature, we estimated the incidence of a character per family, as in the case of animal interactions. We quantified family traits from local species members as these species contributed to the diversity responses we studied. All trait measurements were taken from distinct individuals of each species.

As higher-level features (Fig. 1), we considered the regional species richness of families and, as a proxy of their distribution, the number of plots in which at least one member of the family was surveyed. We also estimated the degree of co-existence among families by means of the C-score[61], an index of reciprocal distribution that ranges from zero, sympatry, to one, allopatry.

**Diversity–environment relationships and data analyses.** Overall, for each family we calculated eight diversity variables for the four main relationships depicted in Fig. 1, which were quantified in presence-only sites for each family. The elevational gradient of community diversity describes changes in species richness at the site level (α-species richness) with increasing elevation. This relationship may be linear or curvilinear, with species richness peaking at some given elevation (Fig. 1 and Supplementary Fig. 3). We used two proxies to describe this relationship in families: the complexity of the species richness-elevation curve and the elevation at which species richness peaks (Peak-α) and its range of variation. The former was expressed in terms of the effective number of parameters or degrees of freedom of the smoothing function (Edf-α) in generalised additive models. As an example, for Edf = 1 the complexity of the curve is approximately the same as that of a linear relationship, and for Edf = 5 complexity approximates a polynomial regression of degree four. Generalised additive models were fitted with a Gaussian distribution of errors on log-transformed species richness vs. elevation with the R package mgcv[62]. In those families that also displayed significant latitudinal and/or longitudinal species richness trends, we fitted models adding geographic coordinates to control for spatial autocorrelation (Supplementary Fig. 3). We built two data sets of families' Edfs, one with estimated Edfs, irrespective of the significance of generalised additive models, and another with corrected Edfs, in which values were set to 0 when models were not significant (i.e., no estimated parameters). Both approximations lead to similar results, the former were reported in the main text and the latter in supplementary material (Supplementary Tables 1 and 5; and Supplementary Fig. 4).

By inspecting curves and richness distribution, and through piecewise regressions of log-species richness vs. elevation, we calculated Peak-α as the curve inflection point[15]. We also derived its standard error from 95% confidence intervals to be used as a measurement error estimate, so that the wider the error, the lower the weight of the associated Peak-α in the following analyses (see below). We used the segmented package of R[63], again controlling for geographic coordinate(s) in families displaying significant latitudinal or longitudinal trends. The other relationships describe changes in species composition, or β-diversity, among sites at increasing geographic distance, elevational distance, or habitat dissimilarity (Fig. 1). The Sørensen index of taxonomic β-diversity was calculated with the function beta in the R package BAT[64] to express pairwise species dissimilarities among sites for each family. Euclidean distances among sites in terms of arcsine transformed cover of habitat types, geographic coordinates and elevation expressed pairwise habitat dissimilarity, geographic distance and elevational distance, respectively. For each family, we regressed pairwise β-diversities among sites on pairwise geographic, elevational and habitat dissimilarities by means of multiple regressions on distance matrices. These multiple regressions were implemented with the MRM function in the R package ecodist[65], with t-values, standard errors of slopes, $r^2$ and significance levels estimated with 999 permutations. We addressed two proxies describing β-diversity-distance or dissimilarity relationships within each family. The first one was the slope of regression (partial coefficient or estimate) to describe species turnover and its range of variation with distance (Slope$_{xy}$), elevation (Slope$_z$) and habitat (Slope$_{hab}$). The standard error of the slope was used as an indicator of measurement error in the subsequent analyses, in order to give less weight to slopes that were poorly estimated in non-significant regressions. The second diversity variable we estimated to describe turnover was the goodness of fit measure $R^2$, which was quantified separately for the three gradients, as a measure of the influence of geographic ($R^2_{xy}$), elevation ($R^2_z$) or habitat ($R^2_{hab}$) dissimilarity on species composition[66]. The coefficient of determination has no associated error term, but non-significant trends ($R^2 = 0$) provide information on the (null) influence of a given predictor on species turnover[40] and can be compared across families (Supplementary Table 3).

obtained from alternative sources, such as the multigene phylogenies of birds[56], insect taxa[57] and lichens[58]; thus, we did not perform analyses using alternative phylogenetic sources. From the literature or our own measurements, we obtained information on functional traits that may critically affect population growth rate, species replacement and establishment, because of their link to mobility, resource use and life histories[15,59,60] (Fig. 1 and Supplementary Note 2). Briefly, for animal families, we estimated the averages of size-corrected wing length, a proxy of mobility, and body size, as a key life-history correlate. In lichens, we quantified growth form type, which ranges from small endolithic leprose lichens to foliose and

Phylogenetic relationships among families within higher taxa were obtained from calibrated phylogenetic trees built from patristic distance matrices with the function *upgma* in the *phangorn*[67] package of R. Prior of comparative analyses on diversity responses, we assessed the power of detecting a phylogenetic signal, as analyses were separately run within lichens, birds and insects with a sample size that ranges from 14 to 9. We used the Monte Carlo-based method for phylogenetic analyses of continuous traits developed by Bottinger et al.[68] and the R package *pmc*[68] for data sets that may contain insufficient power to inform the inference. From the original data sets and phylogenies, we estimated the parameters of two contrasting models: a null model with random draws independent of the phylogeny (white noise WN model) and a simple model of stochastic evolution (Brownian motion BM model) in which changes randomly accumulate over time. Given our small sample, we chose the latter as a model of evolution because it can be described by just two parameters, the starting value of the variable and its evolutionary rate[49]. Moreover, it seems to be the most adequate model for describing the change over time of diversity variables (large number of Blomberg's $K$-values $\approx 1$, Supplementary Table 1). We then computed 1000 simulated data sets from each model at these parameters, and on each data set, the parameters for both models were re-estimated and the likelihood ratio statistic was computed. The collection of likelihood ratio statistics generates one distribution for each model, which in our case are largely overlapping between models in all taxa (Fig. 2). This confirms the low power of tests comparing phylogeny-based vs. null models in our reduced sample size. Yet, when these models were fit to our data set, we were generally able to identify whether BM or WN models better fitted the data (in Fig. 2, dashed lines represent observed $\delta$).

We quantified the phylogenetic signal in α- and β-diversity variables listed above by means of two alternative methods. We estimated the amount of phylogenetic signal by means of Blomberg's $K$ statistics[49] with the *phylosig* function of the *phytools*[69] package of R. Blomberg's $K$ compares the variance of the phylogenetically independent contrasts of the target response against those obtained by reshuffling data across the tree. Stronger deviations from zero indicate stronger relationships between diversity response and phylogeny; $K = 0$ indicates phylogenetic independence and $K = 1$ a phylogenetic pattern compatible with a Brownian motion model of evolution. We included standard errors in analyses of *Peak*-α, *Slope*$_{xy}$, *Slope*$_z$ and *Slope*$_{hab}$ to weight for the strength of diversity responses of families, as larger errors imply weaker and less predictable responses.

As an alternative to Blomberg's statistics, we analysed whether dissimilarities in community responses increased over time by means of regressions on distance matrices. This second method uses, as Blomberg's statistics, randomisations to detect deviations from a null model of phylogenetic independence, but while Blomberg's statistics couples the information of tree topology with independent contrasts, this uses patristic distances —twice node age in phylogeny—as a predictor of effect sizes among families. Effect sizes were expressed by means of Cohen's $d$, a standardised measure of effect (mean difference/pooled standard deviation) for those variables with an associated measure of error. For the estimated degrees of freedom (Edf-α) and coefficients of determination ($R^2_z$), raw differences were calculated. Blomberg's $K$ and regressions on distance matrices were also performed to assess the amount of phylogenetic signal in average organismal trait values, family regional species richness, and the number of occurrence plots of families. We also investigated whether patterns of co-existence among families correlate with their evolutionary relationships, by means of regression on distance matrices of the pairwise $C$-score matrix on patristic distance. We assessed whether families contributing to analyses (i.e. widespread and species-rich ones) were phylogenetically over-dispersed or clustered with respect to the regional pools of each major taxa (36 avian, 31 insect and 33 lichen families with available phylogeny). For this, we tested for departures from random associations with $D$ statistics, a measure of phylogenetic signal in binary traits (widespread vs. rare families), calculated with the *phylo.d* function of the R package *caper*[70].

Finally, we tested for the association of diversity responses with organismal and higher-level features, centring on those diversity responses that displayed the most consistent phylogenetic signal across methods (listed in Table 1). We fitted generalised least square regressions of the R package *nlme*[71] and *ape*[72], weighting data for the variance of diversity responses when available, and adding a correlation structure of a Brownian motion model of trait evolution. We first fitted intercept-only models and then added predictors and the correlation structure in a forward/backward manner. We ranked models on the basis of the Akaike's Information Criterion for small sample size (AICc), assuming that the best models were separated by at least three AICc from the rest of the models (ΔAICc < 3) and had high-AICc weight[73]. With this procedure, we could assess the influence of phylogeny on these relationships in the presence of covariates that may themselves be affected by phylogeny. All statistical tests were two-sided.

**Reporting summary**. Further information on research design is available in the Nature Research Reporting Summary linked to this article.

## Data availability

The raw data set has been deposited in the Digital CSIC repository (https://doi.org/10.20350/digitalCSIC/10529) (ref. [74]). Source data underlying Figs. 2, 3 are provided as Source Data file.

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

## Acknowledgements

We are grateful to A. Segura for help during fieldwork, to A. Anadón for facilitating access to the Oviedo University insect collection, to S. Young for performing a language review, and to Picos de Europa National Park and Principado de Asturias for providing permissions to collect insect specimens. This work was funded by grants CGL2014-53899-P/AEI/FEDER.UE and CGL2017-85191-P/AEI/FEDER.UE of the Spanish Ministry of Science, Innovation and Universities, and grant IDI/2018/000151 of Principado de Asturias.

## Author contributions

P.L., J.R.O., J.P., B.J.-A. collected and reviewed data, and discussed results. P.L. designed the study, performed statistical analyses and wrote the paper. All authors approved the submission.

## Competing interests

The authors declare no competing interests.
