## [Peer Review File · Nature Communications]

Reviewers' Comments:

Reviewer #1:

Remarks to the Author:

In my opinion this manuscript may become a nice paper. Nonetheless, as it is in this version, it has a major flaw. I can't see why analyses were performed at the family level. My concern is rooted in two major points: 1) Analyses were circumscribed to a single region, which implies most biotic responses to environment are likely to be observed at species/population scales, rather than families. 2) Family richness is regionally low (14 lichen, 9 insect and 9 avian families), which makes any phylogenetic analysis hardly interpretable at family level. I suggest to the authors refining the analyses based on a finer phylogenetic resolution, such as genera, or even species. If phylogenetic resolution is a problem, it can be elegantly solved by incorporating phylogenetic uncertainties to analysis. By doing so, you should attain much more robust results.

Reviewer #2:

Remarks to the Author:

please, see the attached pdf

Review for NCOMMS-19-16515 Patterns of species diversity are evolutionarily conserved across space and the environment

The present study investigates phylogenetic signal in biogeographic patterns. Four patterns are investigated (elevational gradient in species richness, spatial turnover, elevational turnover and habitat turnover in species composition). Each of these patterns is described by appropriate statistical metrics, separately for 32 families of lichens, birds, and insects. Having found phylogenetic signal in the metrics across related families, the authors conclude that biogeographic patterns are phylogenetically conserved. They attribute the conservatism to the conservatism in some of the key organismal traits across the studied families (such as body size, wing length, or growth form).

The study presents a good amount of work. The authors compile novel data. Their analyses are extensive and properly executed. The results reveal an interesting pattern of conservatism in the emergent higher-level properties (slope of elevational gradients, spatial turnover, etc.) of entire communities across multiple unrelated taxa (lichens, birds, and insects), based on newly collected data from northern Spain.

That said, the study is lacking conceptually and the argumentation is not built in a compelling manner. I feel that the authors do not capitalize properly on the large body of interesting results, such that their conclusions might come across as weak. More specific comments and suggestions are given below.

1 - Patterns without the process. The amount of work in terms of data collection and analysis is impressive. But the authors might perhaps consider going one step further and provide the readers with clearer guidance as to what their results mean. Indeed, we can calculate many statistical indices, based on the presented data, use them as input in further analyses to calculate more indices. But the readers often like to be given a clear set of biological questions, unresolved hypotheses and conceptual challenges formulated already in the Introduction. The analyses are then just a means to the resolution of these questions, not a task in itself (e.g. testing for phylogenetic signal).

The authors provide some conceptual background, but might consider providing much more on the theory from the beginning (e.g. there has been a great body of theoretical work on the spatial and habitat turnover, on richness gradients within and across taxa) (e.g. Anderson et al. 2011) to clearly state when and why the examined patterns should show phylogenetic conservatism, and when not. For example, it has been demonstrated that richness gradients are often similar in unrelated higher taxa (e.g. Hawkins et al. 2011) but tend to be dissimilar across closely related taxa (hence the opposite of what the authors argue and find) because related taxa often colonize related niches, compete with each other for shared resources, and therefore do not independently produce similar richness gradients (e.g. Graham et al. 2018).

Perhaps the authors could use these previous results to frame their study in a conceptually more compelling manner: it has been hypothesized that related taxa show similar/dissimilar biogeographic patterns, and the authors now collect and use their own new data to test these two mutually opposing hypotheses to resolve the ongoing debate. This is just one possibility but I feel there is much potential to raise the quality of the current study, if the authors decide to strengthen its conceptual framing and, accordingly, the resultant conclusions.

2 - The writing needs to be more concrete. The writing is formally correct, but does not provide the readers with a concrete idea of what is being addressed and how the results should be understood. The most tangibly written is the Methods section. But in most of the text (Abstract, Intro, Discussion), the authors might need to provide much more of the hard information, clear statements on what has been done and what has not. Previous literature needs to be adequately represented (e.g. work on theory and what the theory implies about the patterns and the processes behind them) and clearly analyzed (what follows from previous work, what has not been resolved in terms of questions, rather than in terms of analyses). Otherwise, the readers are left to their own devices to find how many species were analyzed, how many families, what the setup was. You may have a look, for example, at the first two paragraphs above and compare them to the Abstract to see how much more concrete this summary is compared to the actual text used in the manuscript, which seems to lack much of the required detail that is provided only later, toward the end, in the methods. There are also some minor issues that reflect broader conceptual limitations in the argumentation (e.g. phylogeny cannot be seen as hypothesis or explanation for the detected patterns, given that phylogeny is merely a dataset that captures species relatedness; instead, the authors might want to consider and discuss the processes likely generated the phylogeny and the biogeographic patterns, such that from these combined datasets we can then infer the biological processes that produced the observed data). I am sorry I cannot be more encouraging this time, but hope that at least some of the comments above will be useful.

Anderson et al. (2011) Navigating the multiple meanings of β diversity: a roadmap for the practicing ecologist. *Ecology Letters*

Hawkins et al. (2011) Different evolutionary histories underlie congruent species richness gradients of birds and mammals. *Journal of Biogeography*

Graham et al. (2018) Phylogenetic scale in ecology and evolution. *Global Ecology and Biogeography*

Reviewer #3:

Remarks to the Author:

The study of Laiolo et al. (Patterns of species diversity are evolutionary conserved across space and the environment) intends to evaluate the role of conservatism (conservatism as a process) on the observed patterns of species diversity in northern Spain. The authors used occurrence records and functional data for three different taxonomic groups (insects, birds and lichens) and a higher level phylogenetic tree obtained from the TimeTree of Life.

The study addresses an important question in evolutionary ecology and biogeography. The authors used standard protocols in macroecology and community phylogenetics and found significant phylogenetic signal in species richness patterns. Below some comments are provided that may be useful.

Comments:

The main concern with the current manuscript is that fails to provide clear arguments supporting the author's predictions of phylogenetic signal by the simply quantifying the covariation of family-level phylogenies. Under this prediction, explanations for each of the diversity-environment relationships are expected, yet clear explanations for each relationship were lacking (though the authors present graphical representation in the figure 1).

In addition, given that the work is based on higher taxonomic levels, it is important that the novel information contributed by the analyses is clearly stated. The introduction should provide predictions/expectations to explain how higher taxonomic levels really represent the observed diversity patterns at local and regional scales. The authors explain in some detail the effect of evolutionary history, but fail to clarify how deep-time history drives current species diversity patterns.

Another concern is related with the implementation of the phylogenetic signal. Accordingly, the authors assume that closely related species are to some extent functionally similar [P3, L42-43]—i.e., due to shared ancestry—and used this argument to upscale to higher taxonomic levels, such as the family level. Under this argument, observed trait values are proportional to the divergence in evolutionary time, however, trait values do not depend only on the time, but also on the climatic and geological history where clades originate. For example, the study area in this work has experienced a number of glacial periods, and most communities are the consequence of recent colonization events. Thus the conclusion in this regard is difficult to defend by only testing the phylogenetic signal at family level because the presence of individual species in a given region/area have unique combinations of evolutionary constraints and innovations (evolutionary rates) due to legacies from their biogeographic origins and climatic conditions in which species originated and evolved. In addition, the presence of species in a determinate region/area does not imply that species are adapted to the ecological and environmental of that region/area.

Blomberg's K is used to test phylogenetic signal. Blomberg's K tend to overestimate the phylogenetic signal (high rates of Error type I) when are implemented on pseudo-chronograms (as used in this work). In addition, Blomberg's K is sensitive to the number of tips and strong polytomies which have strong influence on the estimation of phylogenetic signal. For example, the sub-phylogeny for bird families present strong polytomies and some families are missing in the phylogeny (e.g., in the occurrence dataset *Tyto alba* is present but the family Tytonidae is missing in the phylogeny). The other sub-phylogenies also present polytomies. Moreover, it is not clear why the authors merged three different taxonomic groups into a single phylogeny given that the three groups have different evolutionary trajectories.

Another important concern is related to the statistical analyses performed. The authors performed a set of analyses, from Generalized additive models (GAM), Regressions of distance matrices (MRM),

and Generalized least square models (GLS), to describe the changes in species richness and species dissimilarity among sites with increasing elevation. From the fitted the models the authors extracted the coefficients as a proxies for changes in alpha and beta diversity with elevation and used these proxies to estimate phylogenetic signal. The concern with this approach is the uncertainty in the fitted models, given that the coefficients of determination for each model are not presented nor the associated P-values (though the authors present Pseudo-R² for the relationships of beta diversity). How do we know if the reported coefficients and their directions have in fact deviated from random expectation?

Another potential issue is the implementation and posterior interpretation of the regression analyses is the spatial autocorrelation. For example, spatial autocorrelation may inflate the type I error rate in correlative analyses, in the analyses performed the authors are not taking into account the spatial autocorrelation. So, it is possible that spatial correlation is affecting your results/conclusions. This can be solved by calculating spatial filters and then use them as predictors in the regressions. Notice that the regression coefficients might change directions when spatial association is taken into account which may influence the statistical significance. If the results change, it will be necessary to reframe the conclusions.

The last concern relates to the model selection procedure. The authors used AIC to select the best-supported model. However, little information is provided about the procedure used in model selection. For example, the authors said "We ranked models on the basis of the Akaike's Information Criterion for small sample size (AICc), assuming that the best models were separated by at least two AICc from the rest of the models ($\Delta AICc < 2$)" [P13, L281-283]. Translated, the authors are saying that the best-fitting models were selected $\Delta AIC \leq 2$, but, what is the probability that the best model is better than the other candidate models? Or, what percentage of the dependent variable is explained by the best model? This is an important consideration given the uncertainty associated in model selection when the parameter estimates are used for further analysis, such as phylogenetic signal.

Reviewer #1 (Remarks to the Author):

REF1.1) In my opinion this manuscript may become a nice paper. Nonetheless, as it is in this version, it has a major flaw. I can't see why analyses were performed at the family level.

AUTHORS 1.1. From an ecological point of view, there were functional reasons to select families, as they largely correspond to ecological guilds. From an evolutionary point of view, families are resolved taxonomically and are monophyletic while genera are not. Other studies used the same taxonomic resolution when performing comparative studies of diversity patterns across clades. In the revised version, this is explained in **LINES 232-238**.

REF1.2) My concern is rooted in two major points: 1) Analyses were circumscribed to a single region, which implies most biotic responses to environment are likely to be observed at species/population scales, rather than families.

AUTHORS 1.2. We think the referee is referring here to evolutionary responses to the environment: adaptations at the local scale, involving individuals and populations (we refer to this aspect when dealing with “aggregated individual traits”), and speciation above the regional scale, involving entire clades (“higher level traits”). We have clarified the differences between ecological and evolutionary aspects in **FIGURE 1** and **LINES 40-48**.

REF1.3) Family richness is regionally low (14 lichen, 9 insect and 9 avian families), which makes any phylogenetic analysis hardly interpretable at family level. I suggest to the authors refining the analyses based on a finer phylogenetic resolution, such as genera, or even species. If phylogenetic resolution is a problem, it can be elegantly solved by incorporating phylogenetic uncertainties to analysis. By doing so, you should attain much more robust results.

AUTHORS 1.3. At the community scale, the sample of species is reduced as only a small portion of the total diversity of a clade appears in a region, and few families end up coexisting within regions. This is indicated in the introduction, **LINES 57-58**.

Please see AUTHORS' reply 1.1 for discussion on the genus resolution. With respect to the species level, our response variables were ensemble-level properties (species richness and turnover), which could not be calculated at the species level.

Reviewer #2 (Remarks to the Author):

REF. 2.1. The present study investigates phylogenetic signal in biogeographic patterns. Four patterns are investigated (elevational gradient in species richness, spatial turnover, elevational turnover and habitat turnover in species composition). Each of these patterns is described by appropriate statistical metrics, separately for 32 families of lichens, birds, and insects. Having found phylogenetic signal in the metrics across related families, the authors conclude that biogeographic patterns are phylogenetically conserved. They attribute the conservatism to the conservatism in some of the key organismal traits across the studied families (such as body size, wing length, or growth form).

The study presents a good amount of work. The authors compile novel data. Their analyses are extensive and properly executed. The results reveal an interesting pattern of conservatism in the emergent higher-level properties (slope of elevational gradients, spatial turnover, etc.) of entire communities across multiple unrelated taxa (lichens, birds, and insects), based on newly collected data from northern Spain.

That said, the study is lacking conceptually and the argumentation is not built in a compelling manner. I feel that the authors do not capitalize properly on the large body of interesting results, such that their conclusions might come across as weak.

AUTHORS 2.1. In order to provide a sounder conceptual framework to this study, we have almost entirely rewritten the Abstract, Introduction, Results and Discussion, as also detailed in the following replies. In particular, we clarified our aims, hypotheses and predictions (LINES 59-81), added a figure to illustrate them (FIGURE 1), and centred on mechanisms both in the Results (LINES 114-137; FIGURE 3) and Discussion (LINES 159-203). We included novel predictors and added novel results (FIGURE 3, TABLE 1). We re-formulated conclusions on the evolutionary, biogeographic and functional relevance of the study (LINES 204-223).

REF. 2.2. More specific comments and suggestions are given below.

1 -Patterns without the process. The amount of work in terms of data collection and analysis is impressive. But the authors might perhaps consider going one step further and provide the readers with clearer guidance as to what their results mean. Indeed, we can calculate many statistical indices, based on the presented data, use them as input in further analyses to calculate more indices. But the readers often like to be given a clear set of biological questions, unresolved hypotheses and conceptual challenges formulated already in the Introduction. The analyses are then just a means to the resolution of these questions, not a task in itself (e.g. testing for phylogenetic signal).

AUTHORS 2.2. We have centred the new version of the Introduction on the potential underlying mechanisms of observed phylogenetic patterns in diversity responses, following the referee's suggestions on the three points she/he mentioned: We began

introducing BIOLOGICAL QUESTIONS as of the first paragraph of the Introduction, LINES 28-33, then passed to UNRESOLVED HYPOTHESES in LINES 35-41 and 48-54. CONCEPTUAL CHALLENGES are reported throughout the last paragraph of the Introduction, LINES 59-82. PREDICTIONS are mentioned in the Introduction (LINES 71-81) and summarised in FIGURE 1.

REF. 2.3. The authors provide some conceptual background, but might consider providing much more on the theory from the beginning (e.g. there has been a great body of theoretical work on the spatial and habitat turnover, on richness gradients within and across taxa) (e.g. **Anderson et al. 2011**) to clearly state when and why the examined patterns should show phylogenetic conservatism, and when not. For example, it has been demonstrated that richness gradients are often similar in unrelated higher taxa (e.g. **Hawkins et al. 2011**) but tend to be dissimilar across closely related taxa (hence the opposite of what the authors argue and find) because related taxa often colonize related niches, compete with each other for shared resources, and therefore do not independently produce similar richness gradients (e.g. **Graham et al. 2018**). Perhaps the authors could use these previous results to frame their study in a conceptually more compelling manner: it has been hypothesized that related taxa show similar/dissimilar biogeographic patterns, and the authors now collect and use their own new data to test these two mutually opposing hypotheses to resolve the ongoing debate. This is just one possibility but I feel there is much potential to raise the quality of the current study, if the authors decide to strengthen its conceptual framing and, accordingly, the resultant conclusions.

AUTHORS 2.3. The suggestions of the reviewer were very useful to structure the new version of the introduction (LINES 31-41). We reported several study cases on the latitudinal gradient of diversity as well as reviews, and cited all the references mentioned above (REFERENCES 10, 12, 26). These studies served to justify hypotheses based on phylogenetic niche conservatism, which were developed further in the text (LINES 74-81) and in FIGURE 1 for the community level.

We found no evidence sufficient to develop a hypothesis on divergence among closely related clades, thus we did not formulate predictions in this direction. Please consider that taxa in our case are families, thus the statement “related taxa often colonize related niches, compete with each other for shared resources” can be inappropriate for interactions at these higher levels (e.g. it is improbable that all members of a family compete with the members of another family for shared resources and partition their space, this should better fit the case of species within a family). In any case, to account for this possibility, we present results of an analysis comparing the distribution patterns of families, to show that closely related families do not partition in space (Methods: LINES 268-270; Results: LINES 122-124). They did not even appear to partition in niche (FIGURE 2: dissimilarities always increase, not decrease, over evolutionary time).

REF. 2.4. -The writing needs to be more concrete. The writing is formally correct, but does not provide the readers with a concrete idea of what is being addressed and how the results should be understood. The most tangibly written is the Methods section. But in most of the text (Abstract, Intro, Discussion), the authors might need to provide much more of the hard information, clear statements on what has been done and what has not. Previous literature needs to be adequately represented (e.g. work on theory and what the theory implies about the patterns and the processes behind them) and clearly analysed (what follows from previous work, what has not been resolved in terms of questions, rather than in terms of analyses). Otherwise, the readers are left to their own devices to find how many species were analyzed, how many families, what the setup was. You may have a look, for example, at the first two paragraphs above and compare them to the Abstract to see how much more concrete this summary is compared to the actual text used in the manuscript, which seems to lack much of the required detail that is provided only later, toward the end, in the methods.

AUTHORS 2.4. We followed these suggestions, and provided more information in the Abstract, Introduction, Results and Discussion. For this, each of these last three sections is 0.5-1 page longer [we passed information of Methods, Tables and Figures to supplementary material to compensate for this extra text]. We specified what studies have addressed questions that are somewhat similar to ours (latitudinal gradient: Introduction: **LINES 31-35**, Discussion: **LINES 158-171**). The sample size is defined in Results, **LINES 84-85**, and in **FIGURE 2**. The abstract was fully rewritten, with the exception of the first sentence (**LINES 4-15**).

REF. 2.5. There are also some minor issues that reflect broader conceptual limitations in the argumentation (e.g. phylogeny cannot be seen as hypothesis or explanation for the detected patterns, given that phylogeny is merely a dataset that captures species relatedness; instead, the authors might want to consider and discuss the processes likely generated the phylogeny and the biogeographic patterns, such that from these combined datasets we can then infer the biological processes that produced the observed data). I am sorry I cannot be more encouraging this time, but hope that at least some of the comments above will be useful.

AUTHORS 2.5. Thanks for this suggestion. We were much more cautious in this revised version when quoting “the influence of phylogeny”. We substitute these kinds of sentences with “correlations”, “covariation”, “associations”, “connection” or “explanations based on” throughout the text (**LINES 4, 66, 72, 76, 79, 81, 123-126, 131, 146, 188**). In this new version our hypotheses are based on the influence of natural history and the population and community processes that certain organismal traits control. We acknowledge that this kind of historical study cannot test hypotheses [neither on patterns nor processes] with hypothetico-deductive models in the way experimental studies do, a flaw that we cannot overcome.

REF. 2.6. Literature suggested.

Anderson et al. (2011) Navigating the multiple meanings of β diversity: a roadmap for the practicing ecologist. *Ecology Letters*

Hawkins et al. (2011) Different evolutionary histories underlie congruent species richness gradients of birds and mammals. *Journal of Biogeography*

Graham et al. (2018) Phylogenetic scale in ecology and evolution. *Global Ecology and Biogeography*

AUTHORS 2.6. Many thanks; these references were cited (Ref 10, 12, 26).

Reviewer #3 (Remarks to the Author):

REF. 3.1. The study of Laiolo et al. (Patterns of species diversity are evolutionary conserved across space and the environment) intends to evaluate the role of conservatism (conservatism as a process) on the observed patterns of species diversity in northern Spain. The authors used occurrence records and functional data for three different taxonomic groups (insects, birds and lichens) and a higher level phylogenetic tree obtained from the TimeTree of Life.

The study addresses an important question in evolutionary ecology and biogeography. The authors used standard protocols in macroecology and community phylogenetics and found significant phylogenetic signal in species richness patterns. Below some comments are provided that may be useful.

Comments:

The main concern with the current manuscript is that fails to provide clear arguments supporting the author's predictions of phylogenetic signal by the simply quantifying the covariation of family-level phylogenies. Under this prediction, explanations for each of the diversity-environment relationships are expected, yet clear explanations for each relationship were lacking (though the authors present graphical representation in the figure 1).

AUTHORS 3.1. We have reformulated and rewritten the Introduction presenting more clearly the rationale that leads us to hypothesise a correlation between community diversity patterns and evolutionary history of higher clades (LINES 25-28; 42-58; and the new FIGURE 1). Please also see also AUTHORS' replies 2.1, 2.2 and 2.3 that complete this reply with references to changes in the text.

REF. 3.2. In addition, given that the work is based on higher taxonomic levels, it is important that the novel information contributed by the analyses is clearly stated. The introduction should provide predictions/expectations to explain how higher taxonomic levels really represent the observed diversity patterns at local and regional scales. The authors explain in some detail the effect of evolutionary history, but fail to clarify how deep-time history drives current species diversity patterns.

AUTHORS 3.2 . We put forward predictions based on potential processes at different levels of the biological organisation (individual adaptations or higher level traits LINES 77-81) that may affect population dynamics and community structure. The new FIGURE 1 reports information on these potential processes. Please also see REPLY 1.2, 2.2, 2.1 and 2.3.

REF. 3.3. Another concern is related with the implementation of the phylogenetic signal. Accordingly, the authors assume that closely related species are to some extent functionally similar [P3, L42-43]—i.e., due to shared ancestry—and used this argument to upscale to higher taxonomic levels, such as the family level. Under this argument, observed trait values are proportional to the divergence in evolutionary time, however,

trait values do not depend only on the time, but also on the climatic and geological history where clades originate. For example, the study area in this work has experienced a number of glacial periods, and most communities are the consequence of recent colonization events. Thus the conclusion in this regard is difficult to defend by only testing the phylogenetic signal at family level because the presence of individual species in a given region/area have unique combinations of evolutionary constraints and innovations (evolutionary rates) due to legacies from their biogeographic origins and climatic conditions in which species originated and evolved.

AUTHORS 3.3. This is a very important point, at the heart of our study system: ecological communities are an ensemble of species originated and evolved elsewhere, which assemble, interact and replace each other at the local scale (Introduction, LINES: 57-58). No study to our knowledge has addressed how phylogenetic niche conservatism can influence patterns that are fully in the ecological domain, how many species coexist in a community and how species replace each other at the local scale. Previous work on conservatism of diversity patterns referred to the distribution of the global diversity of a clade, how it originated and distributed spatially (e.g. across latitudes), thus we understand what the referee is asking and the problem he/she sees. We tried to clarify, since the beginning (THE TITLE) that we were studying conservatism at an ecological scale, of the community, in which species coexist and interact in the same time period and region. The fact that the imprint of evolution is still evident in spite of faunal and floral mixing, of species pools that do not include all members of a clade and have disparate origins, is an important result from our point of view, which we tried to emphasise throughout the text. We now stated clearly that our study refers to the community level: Abstract: LINE 5, Introduction: LINES 42-59, Discussion: LINES 171-176.

REF. 3.4. In addition, the presence of species in a determinate region/area does not imply that species are adapted to the ecological and environmental of that region/area.

AUTHORS 3.4. Individuals of these species have not evolved their adaptations *in situ* but pass local abiotic and biotic filters on the basis of these adaptations. This is now specified in LINES 43-48, 57-58.

REF. 3.5. Blomberg's K is used to test phylogenetic signal. Blomberg's K tend to overestimate the phylogenetic signal (high rates of Error type I) when are implemented on pseudo-chronograms (as used in this work). In addition, Blomberg's K is sensitive to the number of tips and strong polytomies which have strong influence on the estimation of phylogenetic signal.

AUTHORS 3.5. In this revised version, we used three methods to assess phylogenetic patterns, Monte Carlo simulations to qualitatively assess the power of tests, and two quantitative measures of phylogenetic signal, Blomberg K, and regressions on distance matrices (Methods: LINES 326-365 ; Results: LINES 92-113; TABLE S1, FIGURE

2). We only considered as significant results those referring to variables for which all methods identified a significant signal and higher probabilities of Brownian Motion models of evolution with respect to white noise models (LINES 99-100; 110-113) (FIGURE 2). Our sample size is naturally limited by the regional assemblage size. In spite of a huge sampling effort, sample size for tests ranges from 9 to 14, the probability of type II error in these conditions is high (LINES 106-113). Please also see FIGURE 2 depicting density plots of log-likelihood ratio tests. The significance of these tests was not assessed as BM and WN models have the same number of parameters (d.f. = 0) but the position of peaks is informative in this case (in keeping with Ref. 68).

The use of regressions of distance matrices (divergence times, TABLE S3) as an alternative to Blomberg K permitted the analysis of the influence of divergence irrespective of tree topology. That being said, we would like to stress that our trees are chronograms, not pseudo-chronograms, in the sense that no node was placed a posteriori in the lack of phylogenetic information on a family.

REF. 3.6. For example, the sub-phylogeny for bird families present strong polytomies and some families are missing in the phylogeny (e.g., in the occurrence dataset *Tyto alba* is present but the family Tytonidae is missing in the phylogeny). The other sub-phylogenies also present polytomies.

AUTHORS 3.6. Oscine passerines, constituting a large number of families in our sample, radiated in the Eocene (see ref. 37) and the result of this radiation is that divergence occurred in a very short (evolutionary) time, mostly corresponding to 40-46 Ma. Thus, to our knowledge, this is not an artefact due to the lack of phylogenetic information; the phylogeny of oscine birds is based on a large number of studies, much more than in the other two major taxa we studied. Divergence times were similar for most families, but not fully equal, as it appears in the representation of the phylogenetic tree, thus there is an advantage of working with the two types of phylogenetic information. We added this information in LINES 94-96 of Results and LINES 354-363 of Methods.

We analysed diversity patterns in the subset of families in which community patterns could be studied: families had to be species rich and widespread to accurately estimate their diversity variables (to avoid biases in estimates -see also comment 3.8-). In almost 2500 point counts, there were only 9 out of 36 bird families that met the conditions for analyses (outlined in LINES 231; 371-377). Tytonidae owls, mentioned by the referee, are a good example to illustrate this. They include one species only in the study area (and all over Europe), thus community variables (local diversity, beta diversity) could not be calculated for just one species.

REF. 3.7. Moreover, it is not clear why the authors merged three different taxonomic groups into a single phylogeny given that the three groups have different evolutionary trajectories.

AUTHORS 3.7. We clumped families into a single phylogeny for representative purposes only, as now specified in figure captions (FIGURE 2, FIGURE S1). Analyses were performed separately in the three groups. The reason for clumping the three taxa is to illustrate the huge distance between the disparate groups we were working with.

REF. 3.8. Another important concern is related to the statistical analyses performed. The authors performed a set of analyses, from Generalized additive models (GAM), Regressions of distance matrices (MRM), and Generalized least square models (GLS), to describe the changes in species richness and species dissimilarity among sites with increasing elevation. From the fitted the models the authors extracted the coefficients as a proxies for changes in alpha and beta diversity with elevation and used these proxies to estimate phylogenetic signal. The concern with this approach is the uncertainty in the fitted models, given that the coefficients of determination for each model are not presented nor the associated P-values (though the authors present Pseudo-R² for the relationships of beta diversity). How do we know if the reported coefficients and their directions have in fact deviated from random expectation?

AUTHORS 3.8. We accounted for uncertainty in the slopes of regressions and elevation in which species richness peaks, including their variance in analyses since the first version of the manuscript. Non-significant trends have high error terms, and these variables with high errors were weighted in the analyses, in the form of variances (for GLS), standard errors (for Blomberg K) or pooled standard deviations (for effect sizes). These error terms assume the same meaning of intraspecific variation in comparative studies of interspecific trait evolution: highly variable trait values are not removed from analyses; they are given smaller weight (proportional to the magnitude of errors). This information, already present in the Method section and the former Figure 1, is now added in several portions of the text (LINES 295-296; 314-316), as well as in figures or figure captions (FIGURE 1, FIGURE S4).

Edfs and R-squared refer to statistical parameters that are not calculated with error terms. To overcome this problem, in the case of Edfs, we performed analyses with raw Edfs estimated from models, and substituting Edfs values with zero for non-significant generalized additive models (this approach is intuitive as n.s. models have zero parameters) (LINES 288-292); we obtained similar results for both sets of data. Therefore, we decided to present results of the former dataset in the main text, and results of the latter in the supplementary material (TABLE S1; TABLE S7; FIGURE S4).

In the case of the coefficient of determination, we considered that zero values e.g. (the family is not affected by a given type of environmental shift) were informative, as families not affected by the gradient behave similarly with respect to it. Please see also Ref. 40 for a similar approach. We added this information in LINES 320-322.

REF. 3.9. Another potential issue is the implementation and posterior interpretation of the regression analyses is the spatial autocorrelation. For example, spatial

autocorrelation may inflate the type I error rate in correlative analyses, in the analyses performed the authors are not taking into account the spatial autocorrelation. So, it is possible that spatial correlation is affecting your results/conclusions. This can be solved by calculating spatial filters and then use them as predictors in the regressions. Notice that the regression coefficients might change directions when spatial association is taken into account which may influence the statistical significance. If the results change, it will be necessary to reframe the conclusions.

AUTHORS 3.9. Many thanks for this suggestion; we removed embedded spatial patterns in elevation and habitat relationships and present these novel results in place of the former. For estimates of elevation peaks and Edfs, we controlled for latitudinal and longitudinal variation (geographic coordinates) in models when the congeneric community was affected by it and residuals were no more spatially correlated. This information is now added in **LINES 283; 285-287** and **FIGURE S3**. In the case of turnover variables, we calculated partial coefficients in multiple regressions on distance matrices instead of running separate analyses for spatial turnover, elevational turnover and habitat turnover. In this case, we controlled for all gradients at a time. This information is reported in Methods **LINES 308-313** and **Table S5**. Results changed a little only in the case of insects.

We also checked for another kind of spatial autocorrelation, analysing the probability of families to co-occur. We consider that this is the only kind of spatial autocorrelation that may bias our results: families that coexist may exhibit more similar community patterns as they face the same kind of local contingencies in sampling plots. If closely related families tend to coexist, then phylogenetic patterns may appear in the way species assemble and replace across sites. We found no phylogenetic patterns underlying the reciprocal distribution of families: closely related families neither co-occur nor partition in space. This information is reported in **FIGURE 1**, **Methods: LINES 268-270**; **Results: LINES 124**.

REF. 3.10. The last concern relates to the model selection procedure. The authors used AIC to select the best-supported model. However, little information is provided about the procedure used in model selection. For example, the authors said “We ranked models on the basis of the Akaike's Information Criterion for small sample size (AICc), assuming that the best models were separated by at least two AICc from the rest of the models ($\Delta AICc < 2$)” [P13, L281-283]. Translated, the authors are saying that the best-fitting models were selected $\Delta AIC \leq 2$, but, what is the probability that the best model is better than the other candidate models? Or, what percentage of the dependent variable is explained by the best model? This is an important consideration given the uncertainty associated in model selection when the parameter estimates are used for further analysis, such as phylogenetic signal.

AUTHORS 3.10. We followed the referee's suggestion and in this revised version we calculated the probability of each model to be the best one and discussed results based

on the AICc weights (TABLE 1). The procedure of model selection is reported in LINES 125-128 of Results, 378-390 of Methods. Since the probability of a model to be the best one, in some cases, changed a little (e.g. 0.01) when passing to ΔAICc 2 and 3 (while weights were virtually zeros for higher AICc) we used a threshold of three to identify best models (TABLE 1).

Reviewers' Comments:

Reviewer #2:

Remarks to the Author:

Congratulations on a much improved manuscript. As stated in my original assessment, I believe the study has significant potential and develops an important topic, namely the scaling of organismal properties to emergent community-level properties within a phylogenetically explicit framework. The study builds on new data, competently analyzed by state-of-the-art methods, producing an array of empirical results.

I very much appreciate that the authors revised their manuscript thoroughly in line with the suggestions of the three referees (including myself). Still, I feel that further improvements might be possible to fully elaborate and clarify the key ideas of the manuscript. Some suggestions, mostly minor, are given below.

1) Some parts of the manuscript are much stronger than others. Methods and Discussion are very well written and structured. But Abstract and Introduction still seem to lack the necessary detail and focus. As I stated previously, I feel the wording needs to be much more tangible and clear.

Take the following statement from the Abstract: "We analysed whether differences in community responses among higher taxa were associated with their phylogenetic relatedness and constraints to variation in organismal or higher level features. We compared the behaviour of confamilial communities". To the readers, it will not be clear at all what is meant by "community responses" associated with "relatedness and constraints" or by "behavior of confamilial communities".

Instead, it could read something like: "We tested whether community-level patterns (spatial turnover, slopes of the environment-richness relationships) are phylogenetically conserved in the sense their similarity across different organismal families (lichens, insects, birds) is dictated by the relatedness of these families. Correspondence between the community-level patterns and the relatedness of the families that produce the patterns would suggest that phylogenetically conserved properties of organisms translate effectively into the structuring of the entire ecological communities. Consequently, phylogenetic conservatism of community-level patterns informs us about how the historical legacy of the taxon and shared responses among related taxa to similar environments contribute to community assembly". Of course, this is just a quick suggestion and feel free to make any edits, changes, or omissions. But it is simply an example of how the key ideas could be made much more concrete and easy to follow.

2) Introduction should be more structured around Figure 1. There is some redundant information that is not directly related to the topic of the manuscript and could be condensed (first couple of paragraphs). The text becomes more clear and compelling only later on (after line 71). Moreover, there is little mention of emergent patterns, how they are relevant and how they emerge from the properties of individual organisms and species. For these reasons, I feel the Introduction could benefit from further restructuring, condensation of some of the information and from the elaboration of other themes. For example, Figure 1 provides some hypotheses, but these are not sufficiently introduced, motivated, explained and developed in the Intro.

3) Similarly, the Discussion is great. But it lacks a clear conclusion. Perhaps the last paragraph of the Discussion could be revised to leave the readers with a well-formulated take-home message. For example, "related taxa show similar community-level patterns" followed by "this implies..." and explanation how this findings is relevant for further research.

Please, take some of the above suggested formulations as mere examples. I provide these example formulations because I recommended in the previous round to make the text more concrete. While there have been significant improvements, I feel that further work in this direction could be accomplished to maximize the impact of the study. That is why I give examples, to illustrate what approximately I have in mind. However, the actual formulations and changes are, of course, at the full discretion of the authors. In addition, I am aware that the final manuscript will need to balance my comments with those of other reviewers. But I hope it will be possible to include at least some of the comments given above.

Reviewer #4:

Remarks to the Author:

In this manuscript, the authors investigated the role of conservatism on the observed patterns of diversity of three different groups in northern Spain. This paper is a revision of a previously submission in Nature Communications, for which I have acted as an external referee. The paper present findings in a fast-moving area, the understanding of drivers species diversity with the use of phylogenetic approaches. The topic and results have now the potential to interest a broad audience. In addition, the manuscript has received a lot of improvements in the text, the analyses, the methods have been explicated in detail and the discussion have been improved.

The introduction section is now really improved (specially with the setting of the expectations and the figure 1), and I have not more comments. I appreciated you have rephrased many of the former parts including my comments.

Regarding the methods. I am really glad to see that the comments have been included in the main text. I now really like the new organization that is much clearer that the previous submission. As the introduction, I have no more comments.

The results section has also been improved (basically rewritten), whit the clear statements of results. I am also glad to see that the tables have been revised.

The discussion is now very interesting with many former parts removed to this version. I appreciated your organization and the possible explanations for the potential explanations for the observed diversity patterns of these three different groups.

Overall, I want to congratulate the authors for having greatly improved their manuscript. They have taken into account previous comments and suggestions the reviewers have done before.

REVIEWERS' COMMENTS:

Reviewer #2 (Remarks to the Author):

Congratulations on a much improved manuscript. As stated in my original assessment, I believe the study has significant potential and develops an important topic, namely the scaling of organismal properties to emergent community-level properties within a phylogenetically explicit framework. The study builds on new data, competently analyzed by state-of-the-art methods, producing an array of empirical results.

I very much appreciate that the authors revised their manuscript thoroughly in line with the suggestions of the three referees (including myself). Still, I feel that further improvements might be possible to fully elaborate and clarify the key ideas of the manuscript. Some suggestions, mostly minor, are given below.

1) Some parts of the manuscript are much stronger than others. Methods and Discussion are very well written and structured. But Abstract and Introduction still seem to lack the necessary detail and focus. As I stated previously, I feel the wording needs to be much more tangible and clear.

Take the following statement from the Abstract: “We analysed whether differences in community responses among higher taxa were associated with their phylogenetic relatedness and constraints to variation in organismal or higher level features. We compared the behaviour of confamilial communities”. To the readers, it will not be clear at all what is meant by “community responses” associated with “relatedness and constraints” or by “behavior of confamilial communities”.

Instead, it could read something like: “We tested whether community-level patterns (spatial turnover, slopes of the environment-richness relationships) are phylogenetically conserved in the sense their similarity across different organismal families (lichens, insects, birds) is dictated by the relatedness of these families. Correspondence between the community-level patterns and the relatedness of the families that produce the patterns would suggest that phylogenetically conserved properties of organisms translate effectively into the structuring of the entire ecological communities. Consequently, phylogenetic conservatism of community-level patterns informs us about how the historical legacy of the taxon and shared responses among related taxa to similar environments contribute to community assembly”. Of course, this is just a quick suggestion and feel free to make any edits, changes, or omissions. But it is simply an example of how the key ideas could be made much more concrete and easy to follow.

Please note that when mentioning **line numbers**, we refer to the word version with tracked changes in blue [for the pdf versions with changes accepted: line numbers in green].

AUTHORS: We thank the referee for these suggestions, which we used for the abstract (**WORD FILE: Lines 4-17; 13-16; PDF FILE: lines 12-14, 20-22**) and the end of the Introduction (**WORD FILE: Lines 102-105, PDF FILE: lines 81-84**).

2) Introduction should be more structured around Figure 1. There is some redundant information that is not directly related to the topic of the manuscript and could be

condensed (first couple of paragraphs). The text becomes more clear and compelling only later on (after line 71). Moreover, there is little mention of emergent patterns, how they are relevant and how they emerge from the properties of individual organisms and species. For these reasons, I feel the Introduction could benefit from further restructuring, condensation of some of the information and from the elaboration of other themes. For example, Figure 1 provides some hypotheses, but these are not sufficiently introduced, motivated, explained and developed in the Intro.

AUTHORS: We agree with the reviewer. We cut part of the first two paragraphs, in which redundant information was provided (**WORD FILE: Lines 42-48 and 55-63; PDF FILE: lines 38-54**).

We also expanded the text associated to Figure 1 (**WORD FILE: Lines 86-101, PDF FILE: lines 69-80**) as requested, to provide more details on the expected effects of traits on diversity responses.

3) Similarly, the Discussion is great. But it lacks a clear conclusion. Perhaps the last paragraph of the Discussion could be revised to leave the readers with a well-formulated take-home message. For example, “related taxa show similar community-level patterns” followed by “this implies...” and explanation how this findings is relevant for further research.

AUTHORS: We changed the last paragraph of the Discussion, with conclusions, to strengthen this section. We discussed the possible evolutionary and ecological outcomes of this study and how these can inspire future research (**WORD FILE: Lines 236-258; PDF FILE: lines 212-229**).

Please, take some of the above suggested formulations as mere examples. I provide these example formulations because I recommended in the previous round to make the text more concrete. While there have been significant improvements, I feel that further work in this direction could be accomplished to maximize the impact of the study. That is why I give examples, to illustrate what approximately I have in mind. However, the actual formulations and changes are, of course, at the full discretion of the authors. In addition, I am aware that the final manuscript will need to balance my comments with those of other reviewers. But I hope it will be possible to include at least some of the comments given above.

AUTHORS: Many thanks for you thoughtful revision, which has greatly helped to improve the manuscript.

Reviewer #4 (Remarks to the Author):

In this manuscript, the authors investigated the role of conservatism on the observed patterns of diversity of three different groups in northern Spain. This paper is a revision of a previously submission in Nature Communications, for which I have acted as an external referee. The paper present findings in a fast-moving area, the understanding of drivers species diversity with the use of phylogenetic approaches. The topic and results

have now the potential to interest a broad audience. In addition, the manuscript has received a lot of improvements in the text, the analyses, the methods have been explicated in detail and the discussion have been improved.

The introduction section is now really improved (specially with the setting of the expectations and the figure 1), and I have not more comments. I appreciated you have rephrased many of the former parts including my comments.

Regarding the methods. I am really glad to see that the comments have been included in the main text. I now really like the new organization that is much clearer that the previous submission. As the introduction, I have no more comments.

The results section has also been improved (basically rewritten), whit the clear statements of results. I am also glad to see that the tables have been revised.

The discussion is now very interesting with many former parts removed to this version. I appreciated your organization and the possible explanations for the potential explanations for the observed diversity patterns of these three different groups.

Overall, I want to congratulate the authors for having greatly improved their manuscript. They have taken into account previous comments and suggestions the reviewers have done before.

AUTHORS: Many thanks for your comments; we are glad that the new version met your expectations.